# Constrained hybrid modelling to predict microbial dynamics and organic matter turnover in soil systems

Paul Collart [1 2]   Juergen Gall [3 4]   Andrea Schnepf [1 2]   Holger Pagel [1 2]   Lars Doorenbos [3 4]

## Abstract

Soil microorganisms control organic matter cycling and largely determine how soil systems can cope with and mitigate climate change and environmental threats. Representing microbial dynamics in process-based soil models is therefore critical to predict carbon cycling in soils, albeit highly challenging to inform from data. One promising approach to improve their parametrisation is the integration of genomic data, yet modelling the complex and unknown relationship between genomes and the processes the microbes are driving is an unsolved problem. In this work, we present the first hybrid modeling framework for deriving biokinetic parameter values of a process-based soil organic matter turnover model from metagenome-inferred functional traits based on DNA sequencing data. Our model predicts biokinetic parameters of the process-based model from genomic trait data with a neural network and integrates constraints from ecological theory and literature to ensure realistic behavior, even of non-observed state variables. We evaluate our method on synthetic genomic trait datasets of varying complexity and on real data, showing that our approach improves performance over multiple baselines and learns the dynamics of unmeasurable components of the process-based model effectively, even for small training datasets.

## 1. Introduction

Soils store the largest amount of carbon (C) in the terrestrial biosphere, and, as such, understanding soil C dynamics is critical for reliable climate change prediction (Bradford et al., 2016). A key factor of soil C dynamics is soil microorganisms, such as bacteria, that drive the fate of carbon in soils, its storage in soils or its decomposition into carbon dioxide ($CO_2$) in the atmosphere, and are involved in a number of important climate feedback loops (García-Palacios et al., 2021).

To predict soil carbon flows and stocks changes in ecosystems over time, soil process-based models (PBMs) have been developed over time from theoretical frameworks (Manzoni & Porporato, 2009). They are based on a mass balance of conceptual carbon pools (*state variables*) representing soil organic matter fractions with different physicochemical properties (Manzoni & Porporato, 2009). State-of-the-art process-based models integrate the impact of microbial dynamics on carbon utilisation in order to represent the regulation of soil organic C turnover by microbial community composition and activity better than simpler biochemical soil models, which only account for carbon pools (Chandel et al., 2023). Such models integrate microbial physiology (Stolpovsky et al., 2011) and distinguish functional microbial groups based on ecological theory (Pagel et al., 2020).

However, the parametrisation of these mechanistic but more complex models remains a challenge. Biokinetic parameters of the considered functional microbial pools cannot be measured directly, requiring inverse parameter estimation, which is prone to large uncertainty due to model equifinality (Marschmann et al., 2019). Often, only time series of microbial soil respiration ($CO_2$) can be measured and compared with model outputs. Other outputs, such as bacterial biomass, can only be measured at a few time points at best, and represent only the total biomass rather than the abundances of individual microbial pools considered in these models. Genomic data from DNA sequencing (Semenov, 2021) can inform PBMs on potential microbial functions and support their parametrisation (Marschmann et al., 2024). However, despite the increasing availability of genome datasets in soil systems, computational frameworks remain limited in their ability to represent the complex non-linear relationship between functional genes and biogeochemical processes (Guo et al., 2020).

In this work, we propose a hybrid framework that combines

---

[1]Agrosphere (IBG-3), Forschungszentrum Jülich GmbH, Germany [2]Institute of Crop Science and Resource Conservation, University of Bonn, Germany [3]Institute of Computer Science, University of Bonn, Germany [4]Lamarr Institute for Machine Learning and Artificial Intelligence, Germany. Correspondence to: Paul Collart <p.collart@fz-juelich.de>, Lars Doorenbos <doorenbos@iai.uni-bonn.de>.

*Proceedings of the 43$^{rd}$ International Conference on Machine Learning*, Seoul, South Korea. PMLR 306, 2026. Copyright 2026 by the author(s).

a process-based soil model with a neural network to learn the mapping from genomic data to biokinetic parameters of the PBM. This is a novel approach in soil science and has not been explored yet. Specifically, we propose **HySoMi** (HYbrid SOil MIcrobial modelling), a hybrid approach that uses covariates between metagenomic data and state variables or process rates to learn the relation between functional microbial traits and model parameters. Parametrisation of hybrid soil models is particularly challenging due to the equifinality of process-based soil models in combination with the scarcity of datasets that include both metagenome and process measurements. In practice, only time series of microbial soil respiration ($CO_2$) can be measured, whereas the other state variables and the biokinetic parameters, which we aim to learn, cannot be measured. To address this, we further constrain the model to realistic system behaviour of the non-observed state variables and process rates by integrating constraints from ecological theory and other domain knowledge.

We evaluate **HySoMi** on a wide range of experiments on synthetic data, with varying degrees of complexity and dataset size, and on real data. We find that our approach outperforms both unconstrained and non-hybrid approaches across all experiments, and show that **HySoMi** learns the dynamics of unmeasurable components of the model effectively. The success on small datasets, which are common in biogeosciences, further highlights the potential of **HySoMi**. In short, our main contributions are the following:

- We introduce **HySoMi**, a hybrid modelling framework for soil carbon cycling predictions from microbial genomic data.

- We integrate theoretical domain knowledge into **HySoMi** through a constrained loss function to predict realistic microbial dynamics in the absence of measurements.

- We create a synthetic dataset for evaluation and show through a series of experiments that **HySoMi** leads to better performance, even with small training dataset sizes.[1]

## 2. Related Work

Hybrid models seek to combine the benefits of process-based models (PBMs) and machine learning (ML) approaches. While PBMs enable interpretability of interactions due to physics-based equations and the capacity to predict processes under data scarcity, ML-facilitated data-driven approaches allow for representing and discovering complex relationships between system components if a full mechanistic representation is not achievable.

Advances in hybrid model frameworks that combine deep learning with process-based models achieved significant results in fields such as hydrology (Kraft et al., 2022) and ecosystem modelling (Aboelyazeed et al., 2023). Following the differentiable parameter learning framework (Tsai et al., 2021), parameters of the process-based model can be learned from large datasets and related covariates. This approach requires differentiability of the integrated PBM module within the hybrid model, and converting a PBM into a differentiable version of itself can pose a severe challenge. For instance, Aboelyazeed et al., 2023 focused on building a differentiable version of a process-based photosynthesis model. Aboelyazeed et al., 2025 further improved the hybrid model by linking a second neural network that accounts for changes in parameters in response to environmental variables. Kraft et al., 2022 used a neural network to predict a time-varying coefficient of a complex hydrological model from meteorological forcing data. The coefficients are constrained in the loss function to promote near-zero cumulative soil water deficit using self-paced multitask weighting (Kendall et al., 2018).

The integration of PBMs that simulate microbial dynamics and organic matter decomposition in soil systems into hybrid approaches is in its infancy despite their potential use. Xu et al., 2024 combine a simple two-pool soil organic matter model (bacteria and soil organic carbon) and use a Markov chain Monte Carlo sampling algorithm to generate parameter sets to train a neural network generating parameter maps. This approach differs from ours as it trains the neural network directly on estimated parameters, rather than letting the neural network learn the relationship between the used covariates and the PBM parameters.

Reduction of equifinality in models is one of the key challenges in hybrid modelling. The issue of equifinality occurs when different model parameterisations or structures result in equivalent representations of the system (Schmidt et al., 2020), increasing the volume of the behavioural parameter space and reducing the number of parameters that can be efficiently learned by the neural network model (Kraft et al., 2022). ElGhawi et al., 2023 use a hybrid model inferring a single sensitive model parameter, and make use of model regularisation via constraints to reduce model equifinality. They compare two approaches: penalising the loss with an additional loss term (loss regularisation) and inferring an extra auxiliary target variable, where the auxiliary tasks help to regularise the problem objective of inferring sensible heat flux and aerodynamic resistance together. Our contribution builds on this concept, but aims at inferring several process-based model parameters, rather than a single one. We therefore use a more complex loss regularisation approach such that each of the considered model parameters can be learned effectively.

---

[1]The code and dataset are available at `https://jugit.fz-juelich.de/p.collart/hysomi_publication/`

## 3. Method

### 3.1. Problem Setting

We aim to predict microbial dynamics and organic matter turnover from genomic datasets. As genomic data is high-dimensional, and available data is limited, we extract relevant information into a set of $\mathcal{T}$ functional traits using Microtrait (Karaoz & Brodie, 2022), rather than working on the genome data directly. Microtrait uses Metagenome Assembled Genomes (MAGs) as input and a set of rules based on expert knowledge and empirical evidence to derive informative quantitative genomic traits in different categories, such as resource acquisition, resource use, and stress tolerance. These functional traits are grouped into a vector $\mathbf{x} \in \mathbb{R}^{\mathcal{T}}$, where each element describes a property of the underlying genome present in a sample. We aim to learn the mapping from the functional traits to the parameter set $\theta_{bio}$ of the microbial system $g : \mathbb{R}^{\mathcal{T}} \to \Theta$, where $\theta_{bio} \in \Theta$ parametrises a microbial model, which allows us to test hypotheses and better understand and predict the behaviour of the system. However, direct measurements of these parameters are not possible or require extensive experimental work, which does not reflect field conditions.

Instead, we need to rely on the measurable outputs of the system to infer the underlying parameters. While these systems are based on different carbon pools and fluxes that output multiple different observables, which we refer to as *state variables*, $CO_2$ is the only one that can be compared with near-time continuous measurements for model calibration. Other state variables, such as organic carbon and bacterial biomass, can only be measured at best at a few time points, and as a sum of the different pools of the system. As such, each trait vector $\mathbf{x}$ is only associated with a paired $CO_2$ time series $\vec{Y}_{obs} \in \mathbb{R}^t$ of length $t$, representing the measured $CO_2$ emission over time of the system with traits $\mathbf{x}$. Our goal then becomes to learn $g$ from a dataset of $N$ pairs $D = \{(\mathbf{x}_n, \vec{Y}_{n,obs})\}_{n=1}^N$. This task is very challenging, as we cannot observe $\theta_{bio}$ and need to rely on a single state variable.

### 3.2. HySoMi

The setting described above allows for an ML approach to implicitly approximate the mapping $g$ by predicting $\vec{Y}_{obs}$ from $\mathbf{x}$. However, such an approach is unable to predict $\theta_{bio}$, which is required to understand these systems, and ignores the large body of work on modelling microbial systems, which drastically limits the search space by imposing physical constraints. We thus propose the **HySoMi** framework, which integrates PBMs with a data-driven approach to produce more accurate and realistic predictions of microbial dynamics in soils from genome-based data. We first adapt a state-of-the-art soil carbon PBM into a differentiable framework. Then, we use a neural network to predict its

parameters, which parametrise the processes described by the ordinary differential equation system and predict the output state variables. An overview of our method is shown in Fig. 1. We now describe each component.

#### 3.2.1. PROCESS-BASED MODEL

Like many PBMs in soil science, we use a model composed of a set of ordinary differential equations (Manzoni & Porporato, 2009). The PBM uses the predicted biokinetic parameters $\theta_{bio}$ in combination with initial values and fixed physical parameters that are not derived from the genome data ($\theta_{phy}$). The PBM computes time series of state variables ($\vec{Y}_{sim}$). The PBM can be written as:

$$\frac{d\vec{Y}}{dt} = f(\vec{Y}_0, \vec{Y}, \theta_{bio}, \theta_{phy}, t) \tag{1}$$

where $\vec{Y}$ is the state variable vector, $\vec{Y}_0$ the vector of initial values, $\theta_{bio}$ the learnable parameters, and $\theta_{phy}$ the fixed parameters, which are not learned from the input data and related to measured physical soil properties.

We use a simple biogeochemical soil organic matter turnover model. The model accounts for one functional microbial pool but distinguishes between active and dormant organisms (Stolpovsky et al., 2011). The model represents the typical structure of other PBMs, which reflect microbial growth dynamics and physiology, such as Wieder et al., 2015, and calculates time series of six different state variables. The model reflects microbial transformation of high molecular weight organic carbon to low molecular weight organic carbon and the associated microbial growth. The concentration of low molecular weight organic carbon determines microbial dormancy, i.e., the mass-transfer between active and dormant microbial groups. The $CO_2$ is the measurable state variable and represents the cumulative soil respiration. It is the only state variable of the model that can be compared with near-time continuous measurements for model calibration. All model parameters and state variables are described in the Appendix.

To run the PBM during the forward run, we use a differentiable ODE solver (Chen, 2018) that ensures compatibility with gradient descent-based optimisers. The forward integration uses a fith-order Runge-Kutta method, where the gradient of each adaptive step is stored for backward flow. As the ODE solver can be thought of as a series of simple operations, it defines a dynamic computation graph that can be backpropagated through.

#### 3.2.2. PARAMETER ESTIMATION

The goal of our second module is to estimate the PBM parameters from the functional traits. Soil microbial ecology has struggled for many years to integrate with ecosystem-scale biogeochemistry (Schimel, 2023), and very few ex-

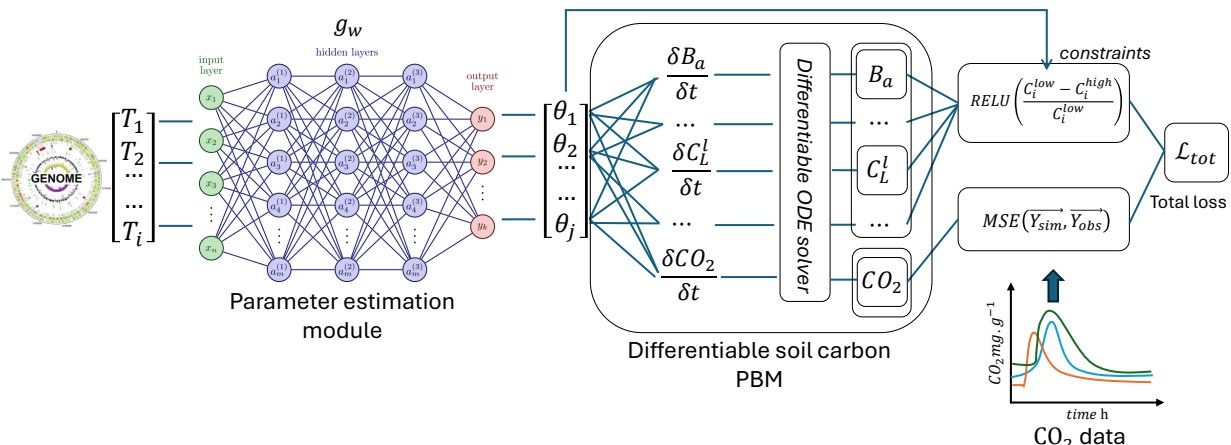

*Figure 1.* **The HySoMi hybrid modelling framework for soil carbon cycling predictions.** The goal is to learn a mapping $g_w$ from genomic data, aggregated into a vector of genomic traits $[\mathcal{T}_1, \mathcal{T}_2, ..., \mathcal{T}_i]$, to biokinetic parameters $\theta_{bio}$. The biokinetic parameters, however, cannot be measured. Instead, time series of $CO_2$ are the only available measurements. To link the unknown biokinetic parameters $\theta_{bio}$ to measurable variables, we utilize a differentiable, process-based carbon soil model (PBM) that takes biokinetic parameters $\theta_{bio}$ as input and predicts state variables, which are time series of carbon pools like active bacterial pool ($B_a$), low molecular weight carbon ($C_L^l$), or carbon dioxide ($CO_2$). Since the state variables as well as $\theta_{bio}$ have physical constraints, such as ranges or logical relationships, we add constraint loss terms for each parameter and state variable and combine them with the mean squared error between the predicted state variables and measured data. The task is very challenging since we do not observe $\theta_{bio}$ or more than one state variable, i.e., except for $CO_2$.

plicit models exist to characterise this relation in its complexity. For this reason, we learn this mapping with a data-driven approach.

Specifically, we map the input traits $\mathbf{x}$ to the PBM parameters $\theta_{bio}$ by learning a model $g_w$, parametrised by weights $w$. We implement $g_w$ as a fully-connected MLP. The parameters of the PBM have different ranges and scales of possible values, making learning more difficult and unstable. To remedy this, we constrain the values using a sigmoid transformation, followed by a projection to the specific parameter ranges

$$\theta_{bio,ranged} = sigmoid(\theta)\,(\theta_{high} - \theta_{low}) + \theta_{low}, \quad (2)$$

where $\theta_{high}$ and $\theta_{low}$ are obtained from literature ranges.

We use the mean-squared error (MSE) between the observed $CO_2$ time series $\vec{Y}_{obs}$ and the $CO_2$ time series obtained from the PBM with the predicted parameters $\vec{Y}_{sim}$ to optimise the network,

$$\mathcal{L}_{MSE} = \frac{1}{t} \sum_{j=1}^{t} \left( \vec{Y}_{sim,j} - \vec{Y}_{obs,j} \right)^2, \quad (3)$$

where $\vec{Y}_j$ denotes the value of $\vec{Y}$ at time step $j$, and $t$ the number of time steps in the time series.

### 3.2.3. LEARNING REALISTIC CONFIGURATIONS

The modules described above can be used to learn the complex mapping between bacterial genomes and PBM parameters using the paired $CO_2$ time series and sequencing data. However, such combined measurements are scarce, and model equifinality further poses a challenge, as only $CO_2$ time series and starting values for total microbial biomass and organic carbon are observable. This can result in converging to a state with non-realistic behaviour for the other, non-observed state variables, which severely limits the interpretability of the outcome. To this end, we design a loss function that integrates the theoretical and expert knowledge into the training by constraining the model to learn the behaviour of unobserved state variables, even when training on a dataset of limited size.

Specifically, we define model constraints in the form of additional loss terms to be minimised during the training process. We consider constraints applied either on parameters of the process-based model $\theta_{bio}$ or on its outputs $\vec{Y}_{sim}$. Both constraint types aim at reducing the model's feasible parameter space by reducing its volume, and constrain the outputs to realistic behaviour for unobservable state variables.

The constraints are defined as inequalities and listed in Table 1. The inequalities can have one variable or two variables. For simplicity, we will discuss the general case:

$$\alpha a < \beta b, \quad (4)$$

where $\alpha$ and $\beta$ are two scalars, and $a$ and $b$ are two variables. For each inequality, we obtain a loss term as:

$$\mathcal{L}_{a,b} = ReLU\left( \frac{\alpha a - \beta b}{\alpha a} \right). \quad (5)$$

*Table 1*. **Constraints used for our loss function and synthetic dataset generation.** We use typical literature values or ecological considerations. All variables are defined in the Appendix. Constraints are labeled by P if they apply to model outputs (state variables), and C if they apply to PBM parameters. $\mathbb{1}(x)$ is the indicator function that is 1 if $x$ is true. $B_{tot}$ and $C_{tot}$ are defined as $B_{tot} = B_a + B_i$, $C_{tot} = C_H + C_L + C_L^s$. $j$ represents time step, with $t$ the last time point of the time series.

| Constraints | Description | Source | Label |
|---|---|---|---|
| $B_{tot,j} < 3.180 \quad \forall j = 1, \ldots, t$ | The total amount of microbial biomass is relatively small (50–3180 $\mu g.g^{-1}$ soil). | Dragone et al. (2024) | P1 |
| $B_{tot,j} > 0.05 \quad \forall j = 1, \ldots, t$ | The total amount of microbial biomass is relatively small (50–3180 $\mu g.g^{-1}$ soil). | Dragone et al. (2024) | P2 |
| $C_{tot,j} < 65 \quad \forall j = 1, \ldots, t$ | Range of total organic matter in soils. | Blanco-Canqui et al. (2013) | P3 |
| $C_{tot,j} > 1.5 \quad \forall j = 1, \ldots, t$ | Range of total organic matter in soils. | Blanco-Canqui et al. (2013) | P4 |
| $\max_j(B_{tot,j}) > 2B_{tot}^{ini}$ | Bacteria population grows and doubles their biomass following a large pulse of glucose. | Endress et al. (2024); Reischke et al. (2014) | P5 |
| $B_{i,t} > 0.5 \max_j(B_{a,j})$ | Most of the maximum bacterial population switches to dormancy when low molecular weight carbon is depleted. | Hobbie & Hobbie (2013) | P6 |
| $\arg\max_j(B_{a,j}) < 48$ | Maximum growth of the bacterial population is reached before 48h after pulse. | Endress et al. (2024) | P7 |
| $\arg\max_j(B_{a,j}) > 15$ | Maximum growth of the bacterial population is reached later than 15h after pulse. | Endress et al. (2024) | P8 |
| $\sum_{j=1}^{t} \mathbb{1}(B_{a,j} > 0.5 \max_j(B_{a,j})) < 48$ | Bacterial population is above 0.5 of its maximum for not more than 48h. | Endress et al. (2024) | P9 |
| $\sum_{j=1}^{t} \mathbb{1}(B_{a,j} > 0.5 \max_j(B_{a,j})) > 5$ | Bacterial population is above 0.5 of its maximum for more than 5h. | Endress et al. (2024) | P10 |
| $C_{H,t} > 0.9 \, C_H^{ini}$ | In the 7-day incubation period, only a small amount of high molecular weight carbon is degraded. | Logical consideration for the system | P11 |
| $m_{max} < \mu_{max}$ | Maintenance rates are lower than growth rates. | Logical consideration for the system | C1 |
| $\mu_{max} < d$ | Deactivation rates are higher than growth rates. | Salazar et al. (2018) | C2 |
| $d < r$ | Reactivation rates are higher than deactivation rates. | Salazar et al. (2018) | C3 |

For time series, some of the constraints include the maximum like $\max(B_{tot,j})$ or the time when the variable achieves its maximum like $\arg\max(B_{a,j})$. In the case of constraints with two variables, we propagate the gradient for both variables $a$ and $b$. For constraints with an argmax, we use a differentiable approximation of the argmax based on the softmax with temperature set to $1e^{-8}$.

To adaptively balance the loss between the MSE (Eq. (3)) and constraint (Eq. (5)) loss terms, we use homoscedastic uncertainty weighting during training (Kendall et al., 2018; Liebel & Körner, 2018). Instead of manually tuning the weights $\lambda_i$ for each loss term, we add the weights as learnable parameters. The total loss is thus given by

$$\mathcal{L} = \sum_{i=1}^{n+1} \frac{1}{2\lambda_i^2} \mathcal{L}_i + \ln(1 + \lambda^2) \qquad (6)$$

for $n$ constraints, where the regularisation term $\ln(1 + \lambda^2)$ eliminates trivial solutions of learning $\lambda \to \infty$. This approach balances the loss function by prioritising the maximum of easier-to-fulfill constraints over harder ones, offering the maximal fulfilment of the defined constraints without compromising the data fitting.

## 4. Synthetic dataset

There is a lack of coupled $CO_2$ and high-quality metagenomic sequencing data for soils. In particular, there is no dataset with biokinetic parameters that we could use for evaluation, as these parameters are, in most cases, unmeasurable. For this reason, we build a synthetic dataset coupling synthetic input data with realistic output state variable time series. To do so, we use Latin hypercube sampling to obtain parameter sets $\theta_{bio}$ from the possible parameter space, which we then filter for realistic model behaviour. Specifically, we simulate a pulse of 1mg of glucose in 1 gram of soil and the following microbial growth response over a period of 7 days. We keep initial conditions and $\theta_{phy}$ constant across the dataset. To sample realistic model behaviour, we use the set of parameters and process inequality constraints defined in Tab. 1. These constraints are built on expert knowledge, literature, and logical considerations for the modelled system in a similar approach as Sırcan et al., 2025 to constrain the model to realistic output ranges and bacterial dynamics behaviour. The PBM is run forward to generate model outputs, and only parameter sets that fulfill all required constraints are retained in the final dataset.

To simulate the complex relationships between process-based parameters $\theta_{bio}$ and related hybrid model input data, we use a randomly initialised neural network, which transforms 13-dimensional $\theta_{bio}$s into a larger 38-dimensional vector $\mathbf{x}$, representing the aggregated genomic data used as input for the hybrid model. We use a fully connected MLP architecture with ReLU activation functions, where the number of hidden layers and neurons defines the complexity of the input data to the PBM parameters $\theta_{bio}$, with larger random MLP models defining more complex input-output relationships. Outputs are transformed into realistic ranges

for this type of data ([5; 200]) with a sigmoid transformation and scaling. The network is randomly initialised using the approach from (He et al., 2015). The default dataset used was generated using a 3-layer deep network.

For all experiments, we generated datasets of 973 input-output pairs. We reserve 57% of the total data for training (554 samples), and 14% for validation (139 samples). The test set is twice the size of the validation set and has 280 samples. We kept the size of the synthetic dataset relatively low, as real-life data for this kind of application is scarce and expensive to obtain, and 973 samples reflects a good-case scenario when assembling a real-world dataset.

## 5. Experiments

We compare **HySoMi** to multiple baselines to assess its performance in predicting realistic behavior.

**Baselines.** We compare **HySoMi** to a hybrid model without constraints (**Unconstrained**) and an unconstrained model trained on every state variable of the model (**All states**). We also compare to a pure ML model that predicts $CO_2$ outputs directly. The *Unconstrained* method only uses the MSE loss $\mathcal{L}_{MSE}$ and serves as a baseline that only trains on realistically available data without using constraints in the loss function. The *All states* model also only uses $\mathcal{L}_{MSE}$ as its loss function, but uses every state variable as training data, constituting an unrealistic reference scenario, since not all state variables can be measured in practice. This serves as a control for the "best possible performance" for the model, where every time point of each state variable can be observed.

**Metrics.** We use four complementary metrics to assess model performance. As we have access to the real PBM parameters $\theta_{bio}^{data}$ from generating the synthetic dataset, we can use the L1 norm between the parameters predicted by the models $g_w(\mathbf{x}) = \theta_{bio}$ and $\theta_{bio}^{data}$, transformed to $[0, 1]$ ranges, to assess the ability of different methods to retrieve the original PBM parameters. Furthermore, we report the MSE for the observed ($CO_2$) time series as $MSE_{obs}$, and the hidden state variables $MSE_{Hidden}$ to measure the capabilities of the model to predict realistic behavior of non-observed microbial pools and other state variables. Finally, we report the sum of constraint penalties (constraints sum) of the model, which is the only metric, along with $MSE_{obs}$, usable in a real data scenario. $MSE$ metrics are reported in $mg.g^{-1}$.

**Implementation details.** For $g_w$, we use a 38-dimensional linear layer for the input, followed by 5 linear hidden layers with interleaved batch normalisation (Ioffe & Szegedy, 2015) and ReLU activations. We used the Adam optimiser (Kingma, 2015) with a learning rate of 0.005 and a multi-step learning rate scheduler: $lr_{epoch} = \gamma lr_{epoch-1}$

every 10 epochs, with gamma set to 0.9. The batch size is set to 256. All baselines use the same settings where applicable.

### 5.1. Main results

We present the main results of **HySoMi** and the different baselines on the test dataset in Tab. 2. Among the hybrid models, **HySoMi** performs better by 0.0416 in $MSE_{Hidden}$ compared to the unconstrained model, despite the PBM equations and parameter ranges already constraining the system implicitly, showing that the constraints reduce $MSE_{Hidden}$ significantly. The fact that the unconstrained model is unable to reduce the $MSE_{Hidden}$ reveals that the model fits the observed $CO_2$ outputs but predicts them for non-realistic reasons, trapped within the high dimensional parameter space (L1 error $\theta_{bio}^{data}$) of the unconstrained PBM. We observe similar trends for the parameter estimation, where our model outperforms the unconstrained baseline by 0.0948, and the constraint satisfaction, where the difference is 1.0921 in favor of **HySoMi**.

The results of **HySoMi** approximate the results of the reference (*All states*) model, where all six state variables are considered observable and used for training. Regarding $MSE_{obs}$ and $MSE_{Hidden}$ performances, **HySoMi** results are lower by 0.0003 and 0.005, respectively, while for the L1 error of $\theta_{PBM}^{data}$ and the constraints sum, results are better than the *All states* approach by 0.0138 and 0.2489, respectively. These results show that the *All states* reference model performs better in the data-driven MSE metrics at the expense of a poorer identification of the PBM parametrisation in comparison to **HySoMi**.

We find that the pure ML model, which directly learns the mapping from genomic traits to $CO_2$, performs similarly in $MSE_{obs}$ as the hybrid models. However, the model contains no domain knowledge and subsequently has no predictions for the other metrics, leading to an uninterpretable system. On the other hand, hybrid models directly integrate the soil model and provide a comprehensive and mechanistic view of the underlying system.

An analysis of the MSE performance with respect to the six individual state variables (Fig. 2) shows that the *Unconstrained* model is unable to predict realistic inactive bacteria ($B_i$) and high molecular weight carbon ($C_H$), explaining most of the difference in $MSE_{Hidden}$. On the other hand, **HySoMi** reaches an MSE close to the *All states* reference model for these two state variables. Overall, **HySoMi** consistently predicts accurate behaviour of the six state variables on the test data including realistic carbon turnover in soils, with realistic active and inactive microbial pools. This is particularly important, as this shows the model can recover realistic parametrisation by reducing the feasible parameter space of the process-based model. Furthermore, the

*Table 2.* **Performance on the test dataset.** We report mean and standard deviation over 6 random seeds for 3 hybrid models: **HySoMi**, **Unconstrained**, and **All states**. Best performance shown in bold. A pure ML model of the same size is added for comparison.

| | $MSE_{obs}$ | $MSE_{Hidden}$ | L1 error $\theta_{PBM}^{data}$ | Constraints sum |
|---|---|---|---|---|
| **HySoMi** | **0.0355** $\pm$ 0.0006 | **0.0211** $\pm$ 0.0018 | **0.2336** $\pm$ 0.0053 | **0.2823** $\pm$ 0.1366 |
| **Unconstrained** | 0.0356 $\pm$ 0.0007 | 0.0627 $\pm$ 0.0090 | 0.3284 $\pm$ 0.0040 | 1.3744 $\pm$ 0.0660 |
| **ML model** | 0.0360 $\pm$ 0.0001 | N/A | N/A | N/A |
| | | *Reference model* | | |
| **All states** | 0.0352 $\pm$ 0.0005 | 0.0161 $\pm$ 0.0003 | 0.2474 $\pm$ 0.0079 | 0.5312 $\pm$ 0.0732 |

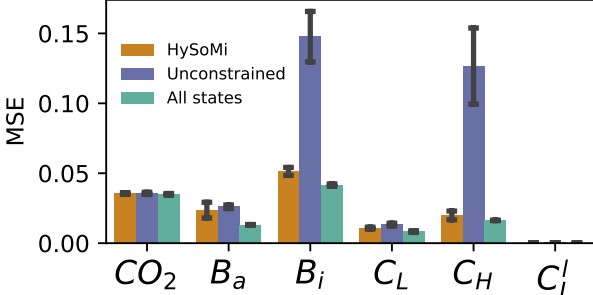

*Figure 2.* **MSE for each state variable on test data.** Both **HySoMi** and *Unconstrained* only use $CO_2$ during training. The *All states* scenario is trained with every state variable. State variables are described in Table A.3. Despite only being trained with $CO_2$, **HySoMi** achieves comparable performance to the unrealistic *All states* model.

hybrid model retains interpretable intermediate variables, which are essential for process-based models.

## 5.2. Effect of dataset complexity and size

To show the generalisability of **HySoMi** to different combinations of data complexity and size, we evaluate the methods on 7 different datasets for which the transformation from input data $\mathcal{T}$ to $\theta_{bio}$ is generated by increasingly deeper randomly initialised networks. We test datasets generated using a randomly initialised MLP with $5, 7, \ldots, 17$ hidden layers and show the results in Fig. 3. Fig. 3A shows that **HySoMi** keeps a consistent behavior across dataset complexities up to data generated with an MLP with 13 layers. For more complex datasets, the predictions worsen, which is expected since the parameter estimation network uses only 5 layers. When increasing the number of layers in the parameter estimation network of the hybrid models (Fig. 3B) as the complexity of the dataset increases, **HySoMi** shows only a slightly degrading performance and $MSE_{Hidden}$ remains low also for more complex datasets.

The training dataset size has little effect on the behavior of our model as shown in Fig. 4. $MSE_{Hidden}$ increases only slightly as the training data size decreases for **HySoMi**. On the other hand, the constraints satisfaction gets worse for the *All states* model as the size of the dataset decreases, due to the training dataset becoming too small to learn constraints

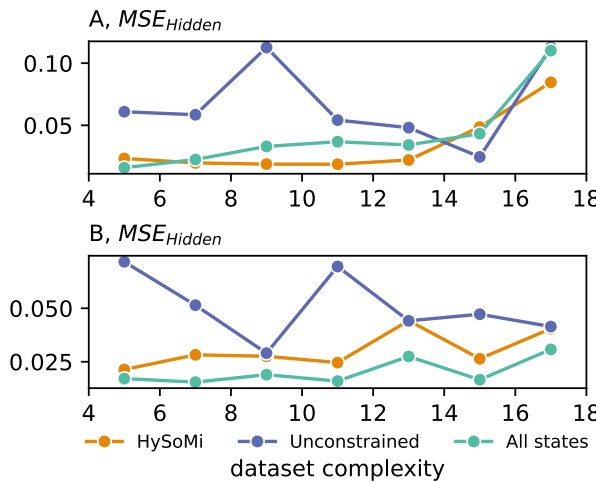

*Figure 3.* **Impact of varying the complexity of the dataset.** **A**: Parameter estimation network $g_w$ of the hybrid module with constant size (5 hidden layers) while increasing the number of layers of the network for generating the dataset from 4 to 18. **B**: Increasing the numbers of layers of the parameter estimation network in the same way as the network for generating the dataset.

directly from the data, even when training on all state variables. In contrast, **HySoMi** keeps the constraint violations low under any dataset size. For the smallest dataset size, **HySoMi** outperforms the *All states* baseline, indicating a better performance under data scarcity. Overall, all tested configurations for **HySoMi** keep consistent performance on smaller datasets due to the use of a hybrid model, leveraging the ability of PBMs to keep accurate predictions even under data scarcity. In the Appendix, we further evaluate the robustness of **HySoMi** to noise in the genomic traits used as input, as well as its sensitivity to its hyperparameters, showcasing similar robustness.

## 5.3. Realism of predicted configurations

Parameter configurations that do not obey the constraints of Tab. 1 have a higher tendency to result in invalid system behaviour. This depends on the nature of the constraints and their impact on the system. To verify that the predicted system behaviour is in line with expectations, we show the average constraint violations on the validation set in Fig. 5. **HySoMi** predictions fulfill all constraints except *P9*, which restricts the active bacteria population to at least 0.5 of its

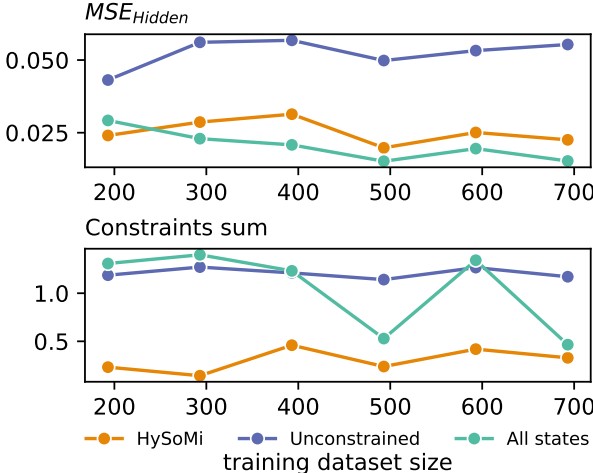

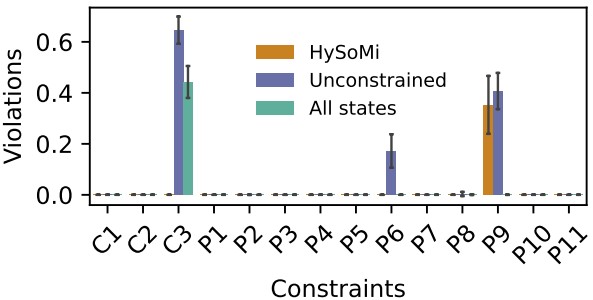

Figure 4. **Impact of dataset size.** $MSE_{Hidden}$ and constraints sum of the different models for different training dataset sizes.

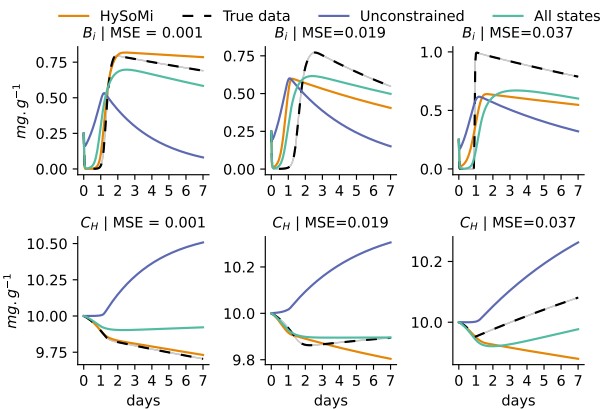

Figure 5. **Constraint satisfaction on validation set after training.** Only a single constraint stays active after training for **HySoMi**. The *Unconstrained* model is not informed by its training data on the P6 and C3 constraints, leading to unrealistic model behavior. Constraints are described in Table 1.

maximum over the time series. The difference in behavior of **HySoMi** compared to the *Unconstrained* approach is explained by the violation of several constraints by the unconstrained model, primarily related to the inactive bacteria pool (*P6*). This is further confirmed by Fig. 2, which shows the *Unconstrained* baseline struggles with predicting the inactive bacterial pool ($B_i$) and high molecular weight carbon ($C_H$).

Fig. 6 shows that **HySoMi** predicts realistic behavior even when $MSE_{hidden}$ for a particular sample is higher than average. Compared to the *Unconstrained* scenario, the model shows realistic behavior for inactive bacteria ($B_i$) in particular. Even when not fitting directly to this state variable, we observe a sharp transition to dormancy and a relatively stable dormant population afterwards, which is expected for the simulated experiment. The *Unconstrained* model is not able to output similar dynamics, and instead shows a fast-decreasing dormant microbial pool. This high mortality for dormant bacteria is unrealistic, as microbial dormancy decreases maintenance needs and reduces decay.

Figure 6. **Prediction of two carbon pools over a 7-day simulation time.** Examples are from 3 different test samples representative of the $MSE_{hidden}$ distribution over samples, with low (left), average (middle), and high (right) MSE. Two different state variables are shown: $B_i$ (Inactive Bacteria) and $C_H$ (High molecular weight carbon).

When comparing the errors of the different models over the simulation time (Fig. C.1), **HySoMi** shows a similar error pattern as *All states* baseline, with only a slow increase over time after the main pulse response. We observe a direct correlation between $MSE_{hidden}$ and the difference in a few parameters $\theta_{bio}$ used by the model. As $MSE_{hidden}$ increases (Fig. 6), errors in parameters related to dormancy ($\zeta$), High molecular weight carbon ($v_{max}, K_H$) and growth ($\mu_{max}$) increase. Less impactful PBM parameters do not show a strong correlation to higher or lower $MSE_{hidden}$. The poor performance of the *Unconstrained* model is linked to a poor estimation of the same parameters, in addition to $\tau$, determining the steepness of the step function controlling dormancy dynamics.

### 5.4. Robustness to constraint misspecification

To analyse the impact of uncertainty in constraint conceptualisation, we validate **HySoMi** under constraint misspecification, i.e., using different constraints for data generation and model training. We generate two additional datasets where P6 and C3 are changed, which are the constraints driving a large part of the differences in performance between the approaches (Fig. 5). For P6, we increased the factor from 0.5 to 0.8, while we modified C3 in two ways, from $d < r$ to $0.7d < r$ and to $0.1d < r$.

We applied the models trained on the original data on these new datasets and show results in Tab 3. For the case with C3 set to $0.7d < r$ and the P6 factor to $0.8$, the results remain very close to the original values. In contrast, when changing C3 to $0.1d < r$, the performance starts to decrease due to a larger misspecification between the constraints used by the model and those of the data. Overall, **HySoMi** is robust

*Table 3.* **Performance of the HySoMi under constraint misspecification.** Despite altering two key constraints, performance remains stable when constraints are reasonably close.

| P6 | C3 | $MSE$ | $MSE_{Hidden}$ | L1 to $\theta_{PBM}^{data}$ |
|-----|-----|--------------------|--------------------|---------------------|
| 0.5 | 1.0 | $0.036 \pm 0.0006$ | $0.021 \pm 0.0018$ | $0.234 \pm 0.005$ |
| 0.8 | 0.7 | $0.039 \pm 0.0014$ | $0.023 \pm 0.0019$ | $0.230 \pm 0.005$ |
| 0.8 | 0.1 | $0.041 \pm 0.0017$ | $0.026 \pm 0.0037$ | $0.299 \pm 0.006$ |

*Table 4.* **Performance with data from the LUCAS soil database (2018).** **HySoMi** provides accurate predictions that are more realistic than the baselines.

|  | $MSE_{obs}$ | Constraints sum |
|-----|--------|---------|
| **HySoMi** | 0.0003 | 0.0006 |
| Unconstrained | 0.0002 | 1.003 |
| ML model | 0.0005 | N/A |

to constraint misspecification as long as the constraints are reasonably close to the data.

### 5.5. Application of HySoMi to real data

Although there are currently no datasets available with deeply sequenced metagenomic data and process rate measurements, we can use limited datasets to provide an initial feasibility study. We present results on data extracted from the European soil database LUCAS (Orgiazzi et al., 2018). This dataset is limited due to shallow DNA sequencing, which prohibits the assembly of MAGs and the inference of traits from metagenomes directly. The original DNA sequences were aligned onto an existing soil MAGs database to allow for trait prediction. The associated $CO_2$ time series comprise only 15h (compared to 7 days in our synthetic dataset), which is too short to observe a full response from the system and represent only the growth phase of the bacterial pool without reaching its maximum peak. We used the data for a first test of our framework with real data, but had to adapt P11 ($C_{H,t} > 0.95\ C_H^{ini}$) and remove constraints P5-P10 that are based on the maximum response of bacterial biomass, which is not reached in this short time series. We designed an additional constraint P12, controlling that the final active bacterial population is larger than 90% of the initial inactive population ($B_{a,t} > 0.9\ B_i^{ini}$), enforcing a dominating effect of the dormancy transition over growth over the small time frame considered.

Tab. 4 shows that all models are able to fit the observed $CO_2$ data. In general, the MSE values are lower than on the synthetic data, as the 15h time series is much shorter and easier to predict. As for the synthetic data, **HySoMi** satisfies the constraints much better than the unconstrained model. For example, C3 (Reactivation rates are higher than deactivation rates) is violated by the unconstrained model on this real data, while **HySoMi** correctly takes this into account.

*Table 5.* **Ablating constraint balancing.** We report results over six seeds on the test set. The homoscedastic uncertainty weighting leads to the best performance.

|  | $MSE_{obs}$ | $MSE_{Hidden}$ | L1 to $\theta_{PBM}^{data}$ | Const. |
|-----|--------|--------|--------|--------|
| **HySoMi** | 0.036 | **0.021** | **0.234** | **0.282** |
| Balanced | 0.035 | 0.030 | 0.243 | 0.710 |
| Normalised | 0.036 | 0.040 | 0.244 | 0.751 |

### 5.6. Ablation study

We compare the adaptive weighting loss of Eq. (6) to a fixed balanced scheme where $\lambda$ is fixed to $1/14$ for each constraint, and the weight of the MSE term of the loss is fixed to 1. Additionally, we compare to a normalised approach, where each term of the loss is normalised to a norm of 1 before multiplying by $\lambda$. The test performance of the adaptive weighting scheme is better than the two other approaches (Tab. 5). Furthermore, individually tuning each $\lambda_i$ becomes increasingly time-consuming as more constraints are added.

## 6. Conclusion

We introduced **HySoMi**, a hybrid modelling framework that combines a neural network with a state-of-the-art soil carbon model to leverage information from metagenomic datasets for soil carbon modelling. **HySoMi** combines information learned from data with theoretical domain knowledge, and we show that our hybrid model can be trained on relatively small datasets common in soil sciences. It is able to learn the dynamics of unmeasurable components better than an unconstrained approach. The study, however, has some limitations. Currently, there is no real dataset with deeply enough sequenced metagenomic data and process rate measurements that can be used to properly evaluate the model. The increasing use of metagenomic data at larger scales, however, will present opportunities to apply this approach for soil science in the near future. Nevertheless, we believe that the newly created dataset will be a challenging benchmark for hybrid models and that the presented results already demonstrate the potential of **HySoMi** for learning a mapping from genomic data to biokinetic parameters using $CO_2$ measurements.

## Impact Statement

This paper presents a hybrid framework to leverage information from genomic datasets to parametrise soil process-based models. As bacteria drive the fate of carbon in soils, this framework could decrease the uncertainty of microbial dynamics in soil models and better predict carbon cycling under climate change.

## Acknowledgments

This work has been funded by the Deutsche Forschungs­gemeinschaft (DFG, German Research Foundation) under Germany's Excellence Strategy, EXC2070 – 390732324 - PhenoRob. We thank Murilo dos Santos Vianna for his help and feedback on code implementation.

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

# A. Process-based model equations

*Table A.1.* Equations of the soil PBM.

| equation | description |
|---|---|
| $$\frac{\delta B_a}{\delta t} = \mu_{max} B_a \frac{K_l C_L^l}{\mu_{max} + K_l C_L^l} - r_d + r_a - \frac{1}{Y_m}\left(r_m^{a,B} - r_m^{a,M}\right) \qquad \text{(A1)}$$ | Active Bacteria |
| $$\frac{\delta B_i}{\delta t} = r_d - r_a - \frac{1}{Y_m}\left(r_m^{d,B} - r_m^{d,M}\right) \qquad \text{(A2)}$$ | Inactive bacteria |
| $$\frac{\delta C_H}{\delta t} = I_H + f_M \overrightarrow{1} \cdot (\overrightarrow{r_B} - \overrightarrow{r_m}) - v_{max} B_a \left(\frac{C_H}{K_H + C_H}\right) \qquad \text{(A3)}$$ | Hight molecular weight organic carbon |
| $$\frac{\delta C_L^l}{\delta t} = I_L + (1 - f_M) \overrightarrow{1} \cdot (\overrightarrow{r_B} - \overrightarrow{r_m}) + v_{max} B_a$$ $$\left(\frac{C_H}{K_H + C_H}\right) - \frac{1}{Y}\mu_{max} B_a \frac{K_l C_L^l}{\mu_{max} + K_l C_L^l} -$$ $$\overrightarrow{1} \cdot \overrightarrow{r_m^M} - k_a C_L^l (C_{max}^s - C_L^s) + k_d C_L^s \qquad \text{(A4)}$$ | Low molecular weight organic carbon |
| $$\frac{\delta C_L^s}{\delta t} = k_a C_L^l (C_{max}^s - C_L^s) - k_d C_L^s \qquad \text{(A5)}$$ | Sorbed low molecular weight organic carbon |
| $$\frac{\delta CO_2}{\delta t} = \frac{1-Y}{Y}\mu_{max} \frac{K_l C_L^l}{\mu_{max} + K_l C_L^l} + \frac{1-Y_m}{Y_m}\overrightarrow{1} \cdot \left(\overrightarrow{r_m^B} - \overrightarrow{r_m^M}\right) + \overrightarrow{1} \cdot \overrightarrow{r_m^M} \qquad \text{(A6)}$$ | $CO2$ |

**Fluxes and functions:**

$r_d$ and $r_a$ are defined as:

$$r_d = \left(1 - \frac{1}{e^{\left(\frac{S_{tr} - C_L^l}{\tau S_{tr}}\right)} + 1}\right) dB_a \qquad r_a = \frac{1}{e^{\left(\frac{S_{tr} - C_L^l}{\tau S_{tr}}\right)} + 1} r B_i \qquad \text{(A7)}$$

Maintenance fluxes are defined as:

$$r_m^{a,B} = m_{max} B_a \qquad r_m^{a,M} = \left(\frac{m_{max} C_L^l k_l}{m_{max} + C_L^l k_l}\right) B_a \qquad \text{(A8)}$$

$$r_m^{i,B} = m_{max} B_i \zeta \qquad r_m^{i,M} = \left(\frac{m_{max} C_L^l k_l}{m_{max} + C_L^l k_l}\right) B_i \zeta \qquad \text{(A9)}$$

$$\overrightarrow{r_m^M} = \begin{pmatrix} r_m^{a,M} \\ r_m^{d,M} \end{pmatrix} \qquad \overrightarrow{r_m^B} = \begin{pmatrix} r_m^{a,B} \\ r_m^{d,B} \end{pmatrix} \qquad \text{(A10)}$$

Model biokinetic parameter $\theta_{bio}$ are defined in Table A.2:

*Table A.2.* Biokinetic parameters of the process-based model. Ranges are obtained from literature.

| Parameter | Interpretation | Units | range | Source |
|---|---|---|---|---|
| $\mu_{max}$ | Maximum growth rate | $d^{-1}$ | $[0.01, 100]$ | (Pagel et al., 2020) |
| $r$ | Reactivation rate | $d^{-1}$ | $[0.01, 100]$ | (Stolpovsky et al., 2011; Pagel et al., 2020) |
| $d$ | Deactivation rate | $d^{-1}$ | $[0.01, 100]$ | (Stolpovsky et al., 2011; Pagel et al., 2020) |
| $Y$ | Growth yield | 1 | $[0.20, 0.995]$ | (Pagel et al., 2020; Manzoni et al., 2012) |
| $K_l$ | Substrate affinity for low molecular weight carbon | $g.mg^{-1}.d^{-1}$ | $[1, 500]$ | (Pagel et al., 2016) |
| $K_H$ | Substrate affinity for high molecular weight carbon | $g.mg^{-1}$ | $[10e^{-6}, 10]$ | (Pagel et al., 2016) |
| $v_{max}$ | Maximum depolymerista-tion rate | $d^{-1}$ | $[1e^{-5}, 1]$ | (Sırcan et al., 2025) |
| $S_{tr}$ | Threshold concentration for inactivation | $mg.g^{-1}$ | $[1e^{-7}, 0.3]$ | (Stolpovsky et al., 2016; Sırcan et al., 2025) |
| $\tau$ | Shape parameter for inac-tivation | 1 | $[1e^{-4}, 1]$ | full range |
| $f_m$ | Distribution factor for bacterial necromass | 1 | $[0.2, 0.9]$ | full range |
| $Y_m$ | Maintenance yield | 1 | $[0.25, 0.995]$ | (Pagel et al., 2020; Manzoni et al., 2012) |
| $m_{max}$ | Maximum maintenance rate | $d^{-1}$ | $[1e^{-4}, 1]$ | (Pagel et al., 2016) |
| $\zeta$ | Reduction factor for dor-mant bacteria | 1 | $[1e^{-4}, 1]$ | full range |

PBM state variables are defined in Table A.3:

*Table A.3.* Each state variable is a time series of length $t$. $B_{tot} = B_a + B_i$ and $C_{tot} = C_H + C_L^l + C_L^s$.

| State variable | Interpretation | Unit |
|---|---|---|
| $B_a$ | Active bacterial pool | $mg.g^{-1}$ |
| $B_i$ | Inactive bacterial pool | $mg.g^{-1}$ |
| $C_H$ | High molecular weight carbon | $mg.g^{-1}$ |
| $C_L^l$ | Low molecular weight carbon | $mg.g^{-1}$ |
| $C_L^s$ | Sorbed phase low molecular weight carbon | $mg.g^{-1}$ |
| $CO_2$ | Carbon dioxyde | $mg.g^{-1}$ |

Initial values for the system are kept the same across samples. Initial bacterial biomass is set to 0.25 mg. As bacteria in bulk soil are dormant under stable conditions, the vast majority of the initial biomass is dormant. $B_a^{ini} = 0.0000125$, $B_i^{ini} = 0.2499875$, $C_L^{l,ini} = 1$, $C_H^{ini} = 10$, $C_L^{s,ini} = 0$, $CO_2^{ini} = 0$. Soil physical parameter are set to: $k_a = 0.1, k_d = 0.01, C_s^{max} = 0.5$ (Kothawala et al., 2009).

## B. Hyperparameter robustness

We evaluate the robustness of our proposed approach with different network dimensions, batch sizes, and learning rates and plot $MSE_{Hidden}$ and constraints sum in Fig. B.1. We tested the network depth from 3 to 16 hidden layers and the hidden dimension from 20 to 90. **HySoMi** keeps consistent behavior across network depth and hidden layer dimensionalities. Furthermore, we test the effect of the batch size for all baselines with values 32, 64, 128, 256, and 512, finding that the behaviour of the different models stays consistent over the different batch size values. We set the default to 256. Finally, we tested the learning rate from 0.001 to 0.03 and did not find a significant effect for the different hybrid models, with **HySoMi** keeping a consistent $MSE_{Hidden}$ over learning rates.

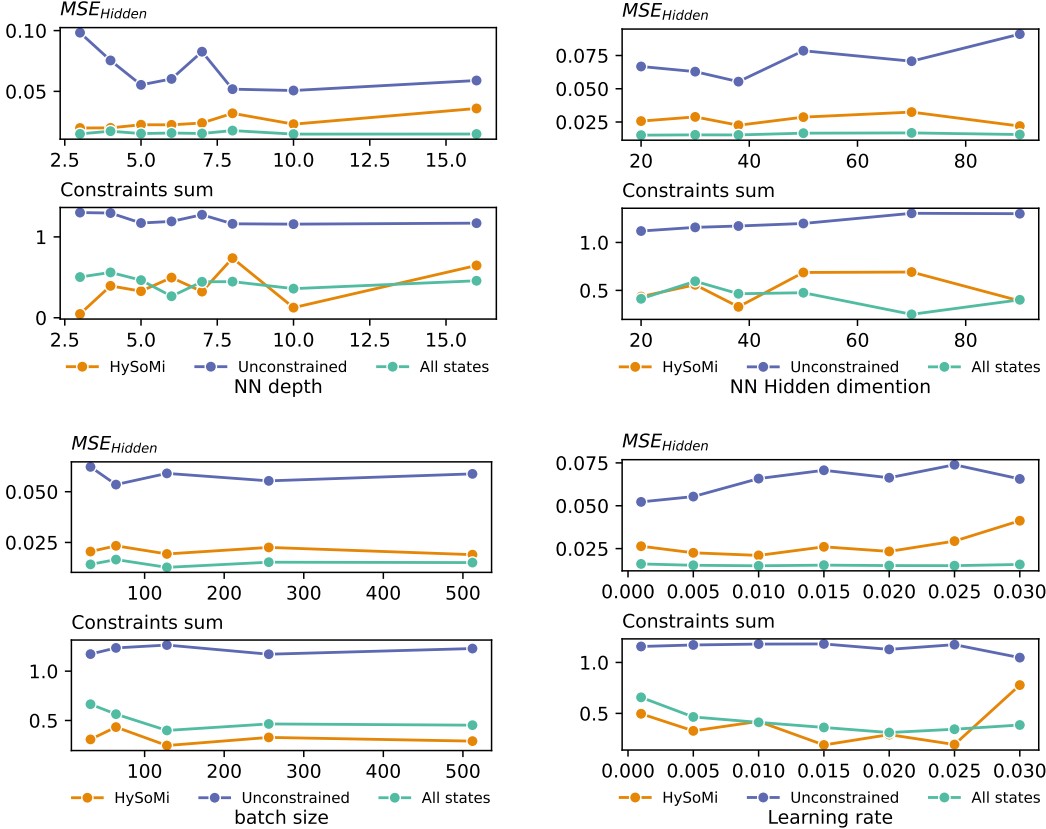

*Figure B.1.* **Hyperparameter sensitivity analysis.** We show the effect of different settings for network depth, hidden layer dimension, batch size, and learning rate for a single run.

## C. Performance over time

In Fig. C.1, we show how the performance of the different models evolves over the simulation time. Also here, **HySoMi** outperforms the unconstrained baseline and behaves similarly to the reference model.

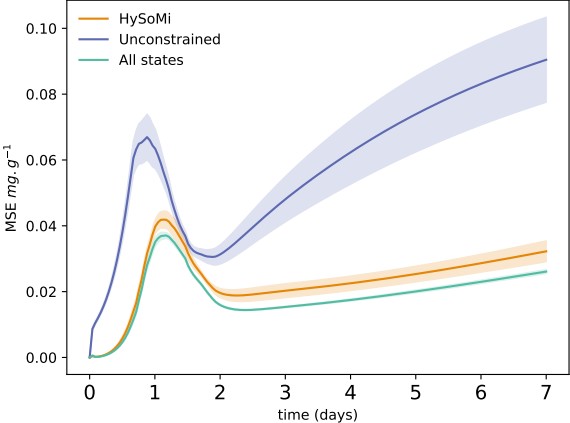

*Figure C.1.* **Mean error of the models on the test dataset as a function of the simulation time.** The total MSE for each time step is summed for the 6 different state variables, with the standard deviation represented as a shaded area. **HySoMi** shows an MSE evolution over time similar to the *All state* reference.

## D. Performance on OOD datasets

### D.1. Robustness to uncertainty in genomic traits

In order to assess the effect of noise on the model, we add Gaussian noise of increasing variance centered around the original input data value to represent uncertainty of the genomic trait values. We show the mean results for **HySoMi** at $\sigma$=3, 6, 10, and 25 (recall the data is in the range [5, 200]) in Tab. D.1. The model shows slight degradation for increasing noise levels, but the results are stable across measured noise levels.

*Table D.1.* **Robustness of HySoMi to noisy inputs.** Performance degrades only slightly and remains robust for noisy inputs.

| $\sigma$ | MSE | $MSE_{Hidden}$ | L1 $\theta_{PBM}^{data}$ | Constraints sum |
|---|---|---|---|---|
| 3 | $0.0353 \pm 0.0006$ | $0.0212 \pm 0.0018$ | $0.233 \pm 0.005$ | $0.280 \pm 0.13$ |
| 6 | $0.0357 \pm 0.0008$ | $0.0212 \pm 0.0018$ | $0.233 \pm 0.005$ | $0.287 \pm 0.13$ |
| 10 | $0.0362 \pm 0.0013$ | $0.0216 \pm 0.0018$ | $0.233 \pm 0.005$ | $0.293 \pm 0.13$ |
| 25 | $0.0375 \pm 0.0337$ | $0.0225 \pm 0.0017$ | $0.233 \pm 0.005$ | $0.318 \pm 0.11$ |

### D.2. Generator-predictor mismatch

We test whether **HySoMi** is also successful when the network used for data generation has a different structure than the one used for prediction. We generated a synthetic dataset using a modified network architecture with 5 hidden layers, a hidden dimension of 42, and tanh activation functions. We show results over a single run in Tab. D.2, where all models achieve slightly worse performance compared to our main results. Still, **HySoMi** fits the observed data well ($MSE = 0.0489$), and greatly reduces the loss on unobserved state variables ($MSE_{Hidden} = 0.0334$) compared to an unconstrained hybrid model trained on the same dataset ($MSE_{Hidden} = 0.0693$). This shows that our approach also works well when the process of generating data uses a very different architecture.

*Table D.2.* **Performance with a mismatch in generator and predictor architecture. HySoMi** is still able to output accurate and realistic predictions.

| Model | $MSE$ | $MSE_{Hidden}$ | L1 to $\theta_{PBM}^{data}$ | Constraints sum |
|---|---|---|---|---|
| **HySoMi** | $0.0489 \pm 0.0091$ | $0.0334 \pm 0.0150$ | $0.2412 \pm 0.0042$ | $0.3483 \pm 0.3245$ |
| Unconstrained | $0.0450 \pm 0.0016$ | $0.0693 \pm 0.0094$ | $0.3195 \pm 0.0129$ | $1.2160 \pm 0.0480$ |
| All states | $0.0263 \pm 0.0015$ | $0.0224 \pm 0.0016$ | $0.2484 \pm 0.0071$ | $0.5046 \pm 0.0876$ |

