# OpenReview forum: "Constrained hybrid modelling to predict microbial dynamics and organic matter turnover in soil systems"
_ICML.cc/2026/Conference — ICML 2026 regular_

### Official Review · Reviewer_u6Dy · 2026-03-05

**Soundness:** 3
**Presentation:** 3
**Significance:** 2
**Originality:** 3
**Overall Recommendation:** 4
**Confidence:** 4

**Summary:**

The paper proposes HySoMi, a hybrid modeling framework designed to predict biokinetic parameters of a process-based soil organic matter model from metagenomic functional traits. Because soil process-based models often suffer from equifinality, the authors incorporate ecological and theoretical constraints into a multi-task loss function to ensure realistic behavior of unobserved state variables. The method is evaluated on a synthetic dataset, demonstrating that theoretical constraints significantly improve the estimation of hidden states compared to unconstrained hybrid models.

**Compliance With Llm Reviewing Policy:**

Affirmed.

**Final Justification:**

Rebuttal addressed main concerns. Soil science and engineering work is solid, hence I maintain the original score.

Personally, I still have concern about the ML for science part after comparing the work with the 2025 application track papers: https://icml.cc/virtual/2025/papers.html?filter=topic&search=Applications->Chemistry,+Physics,+and+Earth+Sciences&layout=topic.

**Key Questions For Authors:**

1. How does the model perform if evaluated on synthetic data that is not pre-filtered strictly by the same constraints used in the loss function?
2. Is any kind of transformation (e.g., logarithmic) used for the hyperparameters? It seems they have very different scale (10^-7~10^3).
3. What is the computational overhead of solving the differentiable ODEs with constraints during the backward pass when scaling to models with multiple functional microbial pools?
4. Microtrait extraction algorithm should be very briefly described. How will the uncertainty and accuracy of the predicted traits impact the overall framework?
5. There are some instances of “dioxyde” that should be dioxide.

**Limitations:**

The authors have discussed the limitations of their work, on the reliance on a synthetic dataset due to the absence of coupled metagenomic and process rate datasets.

**Strengths And Weaknesses:**

Originality: Integrating differentiable biogeochemical models with deep learning to map genomic traits to biokinetic parameters addresses a known bottleneck in soil science.

Significance: Using ecological theory to constrain the loss landscape provides practical utility for scenarios where key state variables cannot be measured continuously. The overall method has potential to be extended to a broad range of engineering problems for dynamical system parameter estimation.

Soundness: It’s good to balance the 14 constraint-based losses via homoscedastic uncertainty weighting that avoids manual tuning.

Weakness:

- Ground-truth data was filtered using the same constraints that form the model's loss regularization, creating a circular validation that could potentially inflate performance metrics.
- The study seems to be more suitable for nature/science series, its alignment with ICML core themes is somewhat limited. An engineering problem of interest is solved, but the performance of the framework/methodology is not assessed on other benchmark problems.
- Figure 5 shows the key results. Although it is challenging to have very accurate predictions for complex dynamical systems, the discrepancy between Hysomi and True data should be discussed.

---

> ### Author Rebuttal · Authors · 2026-03-31
>
> Thank you for the insightful review. We appreciate the points made and address them below.
>
> **Contribution**
>
> We politely disagree with the reviewer that our contribution is limited in the context of Machine Learning for Science, as it tackles an important problem for end-users in soil physics with several technical ML challenges, such as handling both observed and unobserved variables within a hybrid modeling framework. The task of discovering variables that cannot be observed is not only very challenging, but it provides new opportunities to ML researchers for developing novel hybrid models, and this dataset is a very useful setup for evaluation. As also noted by reviewer LQRu, the fact that the originality of our work is focused on an application area is not a significant weakness. Within this application area, our work is highly relevant as it aims to solve the challenging problem in soil science to utilize metagenomic data for robust parameterizations of soil models which enable reliable forecasts with low uncertainty. This is critical for climate models where the uncertainty regarding microbial decomposition is high, or for higher scale soil models that are used to model evolution of soil organic matter.
>
> **Discrepancy between HySoMi and True data**
>
> We want to clarify that Fig. 5 shows qualitative examples of predictions with low, medium, and high MSEs, with the two figures on the right being predictions with the highest MSE in the entire test set. We describe in Section 5.3 that, even in these failure cases, the predictions of HySoMi are more realistic than those of the other models, still predicting reasonable behavior of the system but differing in non-measurable rates. The difference between true data and prediction of HySoMi is dependent on the used constraints, and analysing these failure modes provides a way to build and conceptualise better constraints set.
>
> **Constraint misspecification**
>
> Besides the preliminary experiments on real data shown previously (reviewer 4kzX), for which the exact constraints are unknown, we show results on misspecification robustness in the **Robustness to constraint misspecification** response to reviewer 4kzX.
>
> **Parameter transformations**
>
> We want to clarify that the parameters (Table A.2) in question are not hyperparameters of our method. Instead, they are the output of our network and the input parameters of the PBM. The ranges are from the literature and transformations have not been applied.
>
> **Computational overhead of multiple functional microbial pools**
>
> The typical number of pools stays limited from 1 to 9, with around 50% of existing models having between 2 and 9 microbial pools (Manzoni, 2009), less than an order of magnitude. Nonetheless, extra complexity in the PBM in the form of an extra functional microbial pool should not increase the computational cost excessively due to their limited numbers. The bottleneck for computational cost would still be the forward and backward runs through the ODE system.
>
> **Microtrait**
>
> Microtrait is based on sets of Hidden Markov Models to detect genes and expert knowledge to define traits. We will add a brief description. Note that we do not rely on Microtrait for building our synthetic dataset. Still, to assess the effect of noise on the model, we add Gaussian noise of increasing variance centered around the original input data value. Mean results over 6 runs are shown for HySoMi at sigma=3, 6, 10, and 25. The model shows slight degrading behavior with increasing noise levels, but results are stable across measured noise levels.
>
> | $\sigma^2$ | **MSE** | $MSE_{Hidden}$ | $L1_{\theta}$ | **constraints sum** |
> |--------------|---------|----------------|--------------|---------------------|
> | 3            | 0.0353  | 0.0212         | 0.233        | 0.280               |
> | 6            | 0.0357  | 0.0212         | 0.233        | 0.287               |
> | 10           | 0.036   | 0.0216         | 0.233        | 0.293               |
> | 25           | 0.037   | 0.022          | 0.233        | 0.31                |
>
> **Typos**
>
> Thank you for pointing these out, we will change dioxyde to dioxide wherever needed.

---

> > ### Author Rebuttal · Reviewer_u6Dy · 2026-04-03
> >
> > Thank you for the detailed response. I appreciate the clarifications provided, which have addressed my concerns. I will maintain my positive recommendation.

---

### Official Review · Reviewer_LQRu · 2026-03-08

**Soundness:** 2
**Presentation:** 3
**Significance:** 3
**Originality:** 3
**Overall Recommendation:** 4
**Confidence:** 4

**Summary:**

The paper presents a hybrid modeling framework to predict biokinetic parameters of a process-based soil organic matter turnover model from DNA sequencing data using neural networks. Their framework also incorporate constraints derived from ecological theory into the predictions to allow for an inductive bias to perform well on out of distribution data. The soil matter turnover model is a Process Based Model (PBM) that represents the dynamics as a differential equation model. The total set of parameters in the PBM were lumped into two categories: learnable parameters (meant to be predicted by the NN) and fixed physical parameters (defined from the soil properties). The learnable parameters are predicted by a fully connected MLP followed by a scaling projection. They use a MSE loss function over the simulated CO2 labels, and inequality constraints to penalize solutions that are unreasonable. The multiple loss functions were combined using an adaptive weighting scheme. The framework was tested on a synthetic dataset generated by a randomly initialized NN which maps from parameters to genomic traits. The framework is compared to an unconstrained hybrid baseline, an *all states* upper bound, and a ML model directly predicting CO2 from the genomic traits. The results show the usefulness of adding the constraints into the model as penalty terms.

**Compliance With Llm Reviewing Policy:**

Affirmed.

**Final Justification:**

The authors added the asked experiments: OOD, generator predictor mismatch, and preliminary results on a real world dataset. Due to this, the paper feels more comprehensive and clarifies my concerns. Therefore, I raise my score to 4.

**Key Questions For Authors:**

**Major Question**
1. Some of the constraints have argmax and max functions. How were the gradients propagated for these functions?
2, It might be better to have a very different architecture for the inverse NN. For example, using tanh activation functions for the inverse data generation would yield more non-linearity in the inverse process. That would be a stronger test for the forward NN (with ReLU) to check whether it can actually model the complex non-linear mapping from genomic data to parameters. This test better addresses differences between generator and predictor structure mismatches.
3. Can the authors show some OOD performance based on what is likely to occur in the real world? Ideally HySoMi should outperform the other baselines even under distributional shift. This OOD performance based on a different PBM, constraints, etc provide insight into how well it is likely to perform in the real world. Even addition of some random noise to the measured CO2 or genomic data would help in modeling real-world noise.
4. As the constraints aim to remove the infeasible parameter sets and address the equifinality of the problem, are there chances that even within the feasible set (as defined by the constraints), there could be non-unique parameter solutions? In such cases, how would you identify better solutions? Or are solutions within the parameter set equally likely? In such cases, the L1 on $\theta_{PBM}^{data}$ could becomes unreliable?

**Limitations:**

Yes

**Strengths And Weaknesses:**

**Strengths**

**Soundness**: The addition of constraint losses offer interpretability of results as well addresses the equifinality of the parameters. It improves the prediction of the hidden state variables when compared with the unconstrained hybrid baseline. A good prediction of other state variables could be used in other applications as well as compared with future experimental results.

**Presentation**: The architecture is well structured. All the constraint losses have been well defined with figures and tables.
**Originality**: Based on the provided related work, it seems the work is a novel hybrid model in the field of soil sciences.

It also attempts to create a bridge between genomic data and biogeochemical process models.

**Significance**: The work is well motivated by a lack of rigorous physical models. The framework therefore offers a possible method to predict CO2 from genomic traits of different bacteria. A search method based on this framework could yield engineering applications or better understanding of soil organic matter turnover effects.

The unique perspective could kindle a research into possible patterns that could only be understood from fundamental genomic data.

**Weaknesses**

**Soundness**: The synthetic dataset generation is likely too well suited for a forward prediction. As the authors have utilized a randomly initialized NN to generate the data in an inverse fashion (output -> input), it theoretically lacks the *patterns* that would be found in a real-world dataset. Therefore, the framework probably requires a real-world dataset for testing (even with say 50 samples). It has been mentioned that no real world dataset exists, making such tests quite impossible, but from a technical standpoint good performance on a rigid synthetic dataset doesn't provide enough evidence unfortunately.

Some of the seemingly non-differentiable constraints (argmax, max) lack an explanation in the main text. It is an important aspect of the framework and deserves a small subsection on how they were differentiable-ly enforced..

The paper does lack other baselines (from other work). This could definitely be excused if there are no prior work that attempt this. However, non ML based methods could also be benchmarked against this by adding computational overhead metrics to motivate the use of ML as an efficient approximation.

**Originality**: The paper is original for its application area but not for ML researchers. However, as it is under application track, it is not a significant weakness.

**Significance**: Out-of-distribution (OOD) test set is lacking. As the authors only evaluate on a synthetic dataset, it is quite necessary to test the framework for some OOD performance. The distributional shift could either be created from genome to parameters or constraints. The first type would handle "what happens if the NN sees some new genome data?" while the second handles "some environment conditions have changed the constraints, is my trained NN still useful?".

---

> ### Author Rebuttal · Authors · 2026-03-31
>
> Thank you for the insightful review and for recognising the innovation in our method. We appreciate the points made and address them below.
>
> **Results on real data**
> We acknowledge the importance of validating our method on real data. We refer to the **“Results on real data”** answer to reviewer 4kzX, which describe early results on a real world dataset.
>
> **Other baselines**
> Our work is motivated by a lack of classical approach leveraging sequencing data into soil process based models.  Moreover, there have been no attempts to use machine learning for this task. Therefore, there are no models from prior work that we can compare to.
>
> **Constraint differentiability**
> We use a differentiable implementation of the argmax function based on a softmax with a temperature parameter of 10^8. We use the differentiable PyTorch implementation of the max function, which propagates gradients from the largest element only. We will clarify this in the text.
>
> **Results with generator-predictor mismatch**
> As suggested, we generated a second synthetic dataset using a modified network architecture with 5 hidden layers, a hidden dimension of 42, and tanh activation functions. We show results over a single run in the table below. The models achieve similar relative performance as in our main results, with HySoMi fitting the observed data ($MSE$ = 0.044), and greatly reducing the loss on unobserved state variables ($MSE_{Hidden}$ = 0.021) compared to a non-constrained hybrid model trained on the same dataset ($MSE_{Hidden}$ = 0.069). This shows that our approach also works well when the process of generating data uses a very different architecture. We will include these results using multiple runs.
>
> | **Model**     | **MSE** | $MSE_{hidden}$ | $L1_{\theta}$ | **constraints sum** |
> |---------------|---------|----------------|--------------|---------------------|
> | HySoMi        | 0.044   | 0.021          | 0.246        | 0.168               |
> | Unconstrained | 0.043   | 0.069          | 0.334        | 1.168               |
> | All states    | 0.025   | 0.021          | 0.255        | 0.484               |
>
> **Performance on OOD datasets**
>
> Besides the preliminary experiments on real data shown previously, for which the exact constraints are unknown, we show results on additional OOD datasets. Specifically, we show results for inference on a test dataset generated with constraint misspecification in the **Robustness to constraint misspecification** response to reviewer 4kzX.
> Furthermore, in order to assess the effect of noise on the model, we add Gaussian noise of increasing variance centered around the original input data value. Mean results are shown for HySoMi at sigma=3,6,10 and 25 (with data in range [5, 200]). The model shows slight degrading behavior with increasing noise levels, but results are stable across measured noise levels.
>
> | $\sigma^2$ | **MSE** | $MSE_{hidden}$ | $L1_{\theta}$ | **constraints sum** |
> |--------------|---------|----------------|--------------|---------------------|
> | 3            | 0.0353  | 0.0212         | 0.233        | 0.280               |
> | 6            | 0.0357  | 0.0212         | 0.233        | 0.287               |
> | 10           | 0.036   | 0.0216         | 0.233        | 0.293               |
> | 25           | 0.037   | 0.022          | 0.233        | 0.31                |
>
> **Equifininality**
>
> Indeed, even with the constraints there can still be non-unique solutions in the feasible set. In our synthetic dataset, we generate the ground-truth parameters and the network needs to identify them from the observation. Although other parameters might result in a very similar observation, this ambiguity is in the observation but not in the ground-truth data. We use the L1 and not the L2 norm to avoid that errors due to ambiguities get too much weight. For real data, we do not know the theta values. Using uncertainty modeling of the estimated parameters would be an interesting future work. We will add a discussion to the text.

---

> > ### Author Rebuttal · Reviewer_LQRu · 2026-04-02
> >
> > Thank you for adding OOD results, your new real-world data results, and the data generator mismatch with the predictor. They strengthen the paper
> >
> > I am increasing my score to 4.

---

### Official Review · Reviewer_mcax · 2026-03-11

**Soundness:** 2
**Presentation:** 3
**Significance:** 1
**Originality:** 1
**Overall Recommendation:** 4
**Confidence:** 4

**Summary:**

The paper presents a method for microbial dynamics and organic matter turnover in soil systems based on genomic trades. The method parameterizes an existing microbes process model with an MLP using genomic trades collected from sequencing data. The main novelty seems to be developing an end-to-end trainable parameterisation model.

**Compliance With Llm Reviewing Policy:**

Affirmed.

**Final Justification:**

My original perspective on the application track was that papers published in this conference should bring novelty to the field of Machine Learning even if it is a novel way of applying an existing technique to a particular application. Applying a ML method to an novel application does not fully satisfies this criteria.

Nevertheless, the response of the authors and the rest of the reviews do not share this perspective. Given this, I am adjusting my perspective and recognise this work that advances the state of the art of soil science has a place in the application track of NeurIPS.

The results of the paper clearly show that the method is advancing the state of the art in the field.

**Key Questions For Authors:**

Given the following statement in your related work: ". Xu et al., 2024 combine a simple two-pool soil organic matter model (bacteria and soil organic carbon) and use a Markov chain Monte Carlo sampling algorithm to generate parameter sets to train a neural network generating parameter maps. This approach differs from ours as it trains the neural network directly on estimated parameters, rather than letting the neural network learn the relationship between the used covariates and the PBM parameters."

I assume you are backpropagating through the process model, which is a differential equation. In the method section you do not explicitly describe this process. During training do you backpropagte through the integration of the differential equations given in appendix A? What are the details of this aspect of the training.

**Limitations:**

yes

**Strengths And Weaknesses:**

Even though this paper's primary areas is Application to Chemistry, Physics, and Earth Sciences. The contribution from the perspective of Machine Learning for Science is very limited. In addition the scope of the work in the context of microbial dynamics in soils is very narrow.

The work may have a measurable contribution in the context of its field, but the contribution in the context of Machine Learning or Machine Learning for Science is limited.

---

> ### Author Rebuttal · Authors · 2026-03-31
>
> Thank you for the review. We appreciate the points made and address them below.
>
> **Contribution**
>
> We politely disagree with the reviewer that our contribution is limited in the context of Machine Learning for Science, as it tackles an important problem for end-users in soil physics with several technical ML challenges, such as handling both observed and unobserved variables within a hybrid modeling framework. The task of discovering variables that cannot be observed is not only very challenging, but it provides new opportunities to ML researchers for developing novel hybrid models, and this dataset is a very useful setup for evaluation. As also noted by reviewer LQRu, the fact that the originality of our work is focused on an application area is not a significant weakness. Within this application area, our work is highly relevant as it aims to solve the challenging problem in soil science to utilize metagenomic data for robust parameterizations of soil models which enable reliable forecasts with low uncertainty. This is critical for climate models where the uncertainty regarding microbial decomposition is high, or for higher scale soil models that are used to model evolution of soil organic matter.
>
> **Backpropagation**
>
> We use a differentiable ODE solver to backpropagate through the integration (Section 3.2.1). Specifically, we use the TorchDiffeq library, which implements forward integration and backward gradient flow. Forward integration uses a Runge-Kutta method of order 5, where the gradient of each adaptive step is stored for backward flow. As the ODE solver can be thought of as a series of simple operations, it defines a dynamic computation graph that can be backpropagated through. We will extend the implementation details section with this information.

---

> > ### Author Rebuttal · Reviewer_mcax · 2026-04-03
> >
> > Thank you for your clear response.
> >
> > I recognise that your work has a clear contribution in soil science and I as such I believe it is valuable for this work to be published and reach the target audience. However, I am still convinced that this work would have a bigger impact if it is published in a soil science journal/conference and reviewed by the experts in that field. The paper does not focus sufficiently on discovering hidden variables or developing hybrid models for researchers in other applications to benefit from it. The arguments that you provide on Backpropagation are not significant contribution. On the contrary the progress on differentiable ODE solvers and the developments of Neural ODEs may even bring additional questions up about whether some state of the art developments have been taken into account.
> >
> > That being said, I also recognise your arguments that this is an application track and that this is also the perspective of the rest of the reviews.  Since there is agreement that there is room in this conference for focused application papers, I am happy to change my perspective and see this paper from it's value to the soil science field. As I have indicated at the start, I recognise the novelty of this method in this context, and I will increase my score accordingly.

---

### Official Review · Reviewer_4kzX · 2026-03-13

**Soundness:** 3
**Presentation:** 3
**Significance:** 3
**Originality:** 3
**Overall Recommendation:** 5
**Confidence:** 3

**Summary:**

The paper introduces HySoMi, a constrained hybrid modeling framework that relates metagenome-derived microbial functional traits to biokinetic parameters within a differentiable process-based soil organic matter (SOM) model via a neural network. To mitigate equifinality and ensure ecological plausibility when only CO2 time series are available, the loss function is extended to include inequality constraints based on ecological theory for both parameters and unobserved state variables. HySoMi improves the identifiability of parameters, minimizes constraint violations, and notably improves predictions of unobserved state variables on a range of synthetic data sets of varying complexity and size compared to unconstrained hybrids and machine learning approaches alone, while achieving performance close to a reference model that unrealistically has access to all state variables.

**Compliance With Llm Reviewing Policy:**

Affirmed.

**Final Justification:**

Preliminary findings based on actual data from the LUCAS database (Q2/W1) are useful despite the limited time period used. Robustness tests on constraint misspecification (Q1) indicate that the methodology employed is robust. Authors indicated they will make changes to their equifinality claim (W2).

**Key Questions For Authors:**

1. How robust are the results to constraint misspecification? That is, how do the results for the hidden-state mean squared error and parameter recovery change if the P6 or C3 constraints are changed?
2. Apart from the synthetic random MLP mapping, are there any preliminary results available on the application to actual metagenomic trait data, even if these are small in number, or planned for the case when partial data is available?

**Limitations:**

yes

**Strengths And Weaknesses:**

Strengths:
1. The paper introduces a novel PBM-NN hybrid model that can make multiple biokinetic parameter predictions based on genomic trait covariates rather than relying on post-hoc parameter estimation.
2. The paper proves that the use of constraints can help address equifinality and steer learning towards viable latent dynamics, even when only single-output data (CO2) is available for training the model.
3. The paper also includes ablation studies for constraint weighting strategies and per-state variable errors. The paper evaluates constraint satisfaction during training and also monitors a wide range of metrics to show overall performance trend results.

Weaknesses:
1. The paper only relies on synthetic data for validation. The authors did mention about the limitation, however even a small real dataset could be helpful. The relationship between traits and parameters is based on a randomly initialized MLP that may not reflect the actual relationship between metagenomic traits and processes that can be expected in real-world scenarios.
2. The paper does not include an identifiability analysis to support the claim that equifinality is reduced in the model by demonstrating the recoverability of parameters.

---

> ### Author Rebuttal · Authors · 2026-03-31
>
> Thank you for the insightful review. We appreciate the points made and address them below.
>
> **Results on real data**
>
> We acknowledge the importance of validating our method on real data. However, there are currently no datasets available with deeply sequenced metagenomic data and process rate measurements that can be used to evaluate the model. Still, we are able to present results on a dataset extracted from the LUCAS 2018 European soil database. In this data, the CO2 time series are limited to only 15h (instead of 7 days). Although it is too short to observe a full response from the system, representing only the growth phase of the bacterial pool without reaching its maximum peak being followed by a dormancy transition, we can use it for preliminary results but we needed to adapt (P11) or remove some constraints (P5-P10) that are based on the maximum response, which is not reached in this short time series. For P11 we increased the factor from 0.9 to 0.95 (tighter control over how much of soil organic matter can be depolymerised, since the time series is shorter). An extra constraint was designed (P12), controlling that the final active bacterial population was larger than 90% of the initial inactive population, enforcing a dominating effect of the dormancy transition over growth over the small considered time frame (15h).
>
> We compare the performance of the baselines to HySoMi on this real data in the table below. Note that the “All states” reference model cannot be computed, as we do not have access to the unobserved time series or parameters of the system. Similarly, only the MSE to the observed data and constraints sum can be used as a metric. As in our main results, we find that all models are able to fit the observed CO2 data, with the MSE values being lower than on the synthetic data, as the 15h time series is much shorter. As for the synthetic data, HySoMi satisfies the constraints much better than the unconstrained model. For example, C3 (Reactivation rates are higher than deactivation rates) is violated by the unconstrained model on real data. In general, as deeper metagenomic sequencing becomes more accessible, we expect datasets usable by our hybrid framework to become available and increase in quality and size in the near future, enabling usage and in-depth testing of the framework on real-world high-quality datasets.
>
> | **models**    | **MSE** | **constraint sum** |
> |---------------|---------|--------------------|
> | HySoMi        | 0.0003  | 0.0006             |
> | Unconstrained | 0.0002  | 1.003              |
> | ML model      | 0.0005  | NA                 |
>
> **Equifinality reduction**
>
> We will revise the statement of equifinality in the manuscript. While approaches aiming at analysing and quantifying equifinality directly via parameter identifiability exist (Marschmann, 2019), a full quantitative analysis of equifinality reduction was not possible in the short time. We mainly referred to HySoMi reaching similar MSE values than unconstrained baselines with far lower L1 error to the predicted parameters. We agree that this is not a quantification of equifinality reduction and we will revise the manuscript.
>
> **Robustness to constraint misspecification**
>
> We generated 2 additional test datasets where the constraints used for data generation were shifted, and then applied the pre-trained models on them. As suggested, we change both P6 and C3. For P6, we increased the factor from 0.5 to 0.8. We modified C3 in two ways, to 0.7d<r and to 0.1d<r, simulating the uncertainty in constraint conceptualisation and actual processes.
>
> We present the results over a single run for the experiments in the table below. For the case where 0.7d<r, the results are still close to the original values. In contrast, for 0.1d<r, the metrics start to decrease due to a larger misspecification between the constraints used by the model and those of the data. This shows that the approach is robust to constraint misspecification as long as the constraints are not very far from the data.
>
> | **P6** | **C3** | **MSE** | $MSE_{Hidden}$ | $L1_{\theta}$ |
> |--------|--------|---------|----------------|--------------|
> | 0.5    | 1.0    | 0.035   | 0.021          | 0.233        |
> | 0.8    | 0.7    | 0.039   | 0.023          | 0.230        |
> | 0.8    | 0.1    | 0.041   | 0.026          | 0.299        |

---

> > ### Author Rebuttal · Reviewer_4kzX · 2026-04-04
> >
> > The preliminary findings based on actual data from the LUCAS database (Q2/W1) are useful despite the limited time period used. The robustness tests on constraint misspecification (Q1) indicate that the methodology employed is robust. Authors also indicated they will make changes to their equifinality claim (W2). I raise my score to 5.

---

### Decision · Program_Chairs · 2026-04-30

**Decision:**

Accept (regular)

**Comment:**

This manuscript introduces a hybrid modeling approach to an application from soil science.

The reception to this work during the initial review stage was positive, with reviewers praising the clarity of the presentation, the quality of and strong motivation for the proposed approach, the approach to modeling and problem formulation, and design and analysis of the empirical study. Although the reviewers did point out some relatively minor concerns, these were adequately addressed during the discussion phase. Following the discussion phase, there was universal agreement among the reviewers that the paper was a solid contribution to the conference, and there was universal recommendation of acceptance.

I encourage the authors to take the feedback from the reviewers into account while revising the manuscript.